# Rodent trapping studies as an overlooked information source for understanding endemic and novel zoonotic spillover

**David Simons** [1,2,3]*, **Lauren A. Attfield**[3,4], **Kate E. Jones**[3], **Deborah Watson-Jones**[2,5], **Richard Kock**[1]

**1** Centre for Emerging, Endemic and Exotic Diseases, The Royal Veterinary College, London, United Kingdom, **2** Clinical Research Department, London School of Hygiene and Tropical Medicine, London, United Kingdom, **3** Centre for Biodiversity and Environment Research, University College London, London, United Kingdom, **4** Department of Life Sciences, Imperial College London, London, United Kingdom, **5** Mwanza Intervention Trials Unit, National Institute for Medical Research, Mwanza, Tanzania

* dsimons19@rvc.ac.uk

**Data Availability Statement:** All data and code to reproduce this analysis is available in an archived Zenodo repository (DOI: https://doi.org/10.5281/zenodo.7416054) [32].

## Abstract

Rodents, a diverse, globally distributed and ecologically important order of mammals are nevertheless important reservoirs of known and novel zoonotic pathogens. Ongoing anthropogenic land use change is altering these species' abundance and distribution, which among zoonotic host species may increase the risk of zoonoses spillover events. A better understanding of the current distribution of rodent species is required to guide attempts to mitigate against potentially increased zoonotic disease hazard and risk. However, available species distribution and host-pathogen association datasets (e.g. IUCN, GBIF, CLOVER) are often taxonomically and spatially biased. Here, we synthesise data from West Africa from 127 rodent trapping studies, published between 1964–2022, as an additional source of information to characterise the range and presence of rodent species and identify the subgroup of species that are potential or known pathogen hosts. We identify that these rodent trapping studies, although biased towards human dominated landscapes across West Africa, can usefully complement current rodent species distribution datasets and we calculate the discrepancies between these datasets. For five regionally important zoonotic pathogens (Arenaviridae spp., Borrelia spp., *Lassa mammarenavirus*, Leptospira spp. and *Toxoplasma gondii*), we identify host-pathogen associations that have not been previously reported in host-association datasets. Finally, for these five pathogen groups, we find that the proportion of a rodent hosts range that have been sampled remains small with geographic clustering. A priority should be to sample rodent hosts across a greater geographic range to better characterise current and future risk of zoonotic spillover events. In the interim, studies of spatial pathogen risk informed by rodent distributions must incorporate a measure of the current sampling biases. The current synthesis of contextually rich rodent trapping data enriches available information from IUCN, GBIF and CLOVER which can support a more complete understanding of the hazard of zoonotic spillover events.

**Funding:** D.S. is supported by a PhD award from the UK Biotechnology and Biological Sciences Research Council [BB/M009513/1]. L.A.A. was funded by a PhD award from the QMEE CDT, funded by NERC grant number [NE/P012345/1]. K.E.J is supported by the Ecosystem Services for Poverty Alleviation Programme, Dynamic Drivers of Disease in Africa Consortium, NERC grant number [NE-J001570-1]. D.W-J. receives support from the PREVAC-UP, EDCTP2 programme supported by the European Union [RIA2017S-2014]. D.S. and R.K. are members of the Pan-African Network on Emerging and Re-emerging Infections (PANDORA-ID-NET) funded by the European and Developing Countries Clinical Trials Partnership the EU Horizon 2020 Framework Programme for Research and Innovation [RIA2016E-1609].

**Competing interests:** The authors declare no competing interests.

## Author summary

Emerging and endemic zoonotic diseases are projected to have increasing health impacts, particularly under changing climate and land-use scenarios. Rodents, an ecologically vital order of mammals carry a disproportionate number of zoonotic pathogens and are abundant across West Africa. Prior modelling studies rely on large, consolidated data sources which do not incorporate high resolution spatial and temporal data from rodent trapping studies. Here, we synthesise these studies to quantify the bias in the sampling of rodent hosts and their pathogens across West Africa. We find that rodent trapping studies are complementary to these datasets and can provide additional, high-resolution data on the distribution of hosts and their pathogens. Further, rodent trapping studies have identified additional potential host-pathogen associations than are recorded in consolidated host-pathogen association datasets. This can help to understand the risk of zoonotic diseases based on host distributions. Finally, we quantify the current extent of known rodent presence and pathogen sampling within a species range, highlighting that current knowledge is limited across much of the region. We hope that this will support work to study rodent hosts and their pathogens in currently under sampled regions to better understand the risk of emerging and endemic zoonoses in West Africa.

## 1. Introduction

There is increasing awareness of the global health and economic impacts of novel zoonotic pathogen spillover, driven by the ongoing SARS-CoV-2 pandemic and previous HIV/AIDs and Spanish Influenza pandemics [1]. The number of zoonotic disease spillover events and the frequency of the emergence of novel zoonotic pathogens from rodents are predicted to increase under intensifying anthropogenic pressure driven by increased human populations, urbanisation, intensification of agriculture and climate change leading to altered rodent species assemblages [2–5]. The impact of endemic zoonoses meanwhile remains underestimated [6]. Endemic zoonoses disproportionally affect those in the poorest sections of society, those living in close contact with their animals and those with limited access to healthcare [7–9].

Rodents along with bats contribute the greatest number of predicted novel zoonotic pathogens and known endemic zoonoses [10,11]. Of 2,220 extant rodent species, 244 (10.7%) are described as reservoirs of 85 zoonotic pathogens [10]. Most rodent species do not provide a direct risk to human health and all species provide important and beneficial ecosystem services including pest regulation and seed dispersal [12]. Increasing risks of zoonotic spillover events are driven by human actions rather than by rodents, for example, invasive rodent species being introduced to novel ranges through human transport routes. Rodents typically demonstrate "fast" life histories which allow them to exploit opportunities provided by anthropogenic disturbance [13]. Within rodents, species level traits such as early maturation and short gestation times are associated with increased probabilities of being zoonotic reservoirs [10,14]. Rodent species with these traits are able to thrive in human dominated landscapes, displacing species less likely to be reservoirs of zoonotic pathogens [15]. The widespread occurrence of reservoir species and their proximity to human activity make the description of rodent species assemblages and host-pathogen associations vitally important to understanding the hazard of zoonotic disease spillover and novel zoonotic pathogen emergence [16].

Despite the importance of understanding these complex systems, current evidence on host-pathogen associations is considerably affected by taxonomic and geographical sampling biases [11,17]. Curated biodiversity datasets such as the Global Biodiversity Information Facility

(GBIF) and resources produced by the International Union for Conservation of Nature (IUCN) suffer from well described spatial and temporal sampling biases [18,19]. These data are typically obtained from museum specimen collections and non-governmental organisation surveys. These sampling biases can importantly distort produced species distribution models that are used to infer risk of zoonotic disease spillover [20]. Datasets on host-pathogen associations (i.e., CLOVER) also can suffer from biases introduced from literature selection criteria and taxonomic discrepancies resulting in differential likelihood of accurate host-pathogen attribution by host species. These biases are important because identification of potential geographic hotspots of zoonotic disease spillover and novel pathogen emergence are often produced from these types of host species distributions and host-pathogen associations [21,22]. For example, systematically increased sampling, over-representation of certain habitats and clustering around areas of high human population could lead to an apparent association between locations and hazard that is driven by these factors rather than underlying host-pathogen associations [11,23,24]. Predictions of zoonotic disease spillover and novel zoonotic pathogen emergence must account for these biases to understand the future hazard of zoonotic diseases [22].

West Africa has been identified as a region at increased risk for rodent-borne zoonotic disease spillover events, the probability of these events are predicted to increase under different projected future land-use change scenarios [4,25]. Currently within West Africa, some rodent species are known to be involved in the transmission of multiple endemic zoonoses with large burdens on human health, these pathogens include Lassa fever, Schistosomiasis, Leptospirosis and Toxoplasmosis [26,27]. The presence of other species within shared habitats may mitigate the spread of these pathogens through the "dilution effect", where ongoing loss of biodiversity may further increase the risk to human populations [5]. Understanding of the distribution of these zoonoses are limited by biases in consolidated datasets. Rodent trapping studies provide contextually rich information on when, where and under what conditions rodents were trapped, potentially enriching consolidated datasets [28]. Studies have been conducted in West Africa to investigate the distribution of rodent species, their species assemblages, the prevalence of endemic zoonoses within rodent hosts (e.g., Lassa fever, Schistosomiasis) and to identify emerging and novel zoonotic pathogens [29–31]. However, individual level data from these studies have not previously been synthesised for inclusion in assessments of zoonotic disease spillover and novel zoonotic pathogen emergence.

Here, we synthesise rodent trapping studies conducted across West Africa published between 1964–2022. First, we use this dataset to investigate the geographic sampling biases in relation to human population density and land use classification. Second, we compare this to curated host datasets (IUCN and GBIF) to understand differences in reported host geographic distributions. Third, we compare identified host-pathogen associations with a consolidated dataset (CLOVER) to explore discrepancies in rodent host-pathogen associations and report the proportion of positive assays for pathogens of interest. Finally, within our dataset we investigate the spatial extent of current host-pathogen sampling to identify areas of sparse sampling of pathogens within their host ranges. We expect that rodent trapping studies provide an important additional source of high-resolution data that can be used to enrich available consolidated datasets to better understand the hazard of zoonotic disease spillover and novel zoonotic pathogen emergence across West Africa.

## 2. Methods

### 2.1. Data sources

**2.1.1. Host and pathogen trapping data.**    To identify relevant literature, we conducted a search in Ovid MEDLINE, Web of Science (Core collection and Zoological Record), JSTOR,

BioOne, African Journals Online, Global Health and the pre-print servers, BioRxiv and EcoE-voRxiv for the following terms as exploded keywords: (1) Rodent OR Rodent trap* AND (2) West Africa, no date limits were set. We also searched other resources including the UN Official Documents System, Open Grey, AGRIS FAO and Google Scholar using combinations of the above terms. Searches were run on 2022-05-01, and returned studies conducted between 1964–2021.

We included studies for further analysis if they met all of the following inclusion criteria; i) reported findings from trapping studies where the target was a small mammal, ii) described the type of trap used or the length of trapping activity or the location of the trapping activity, iii) included trapping activity from at least one West African country, iv) recorded the genus or species of trapped individuals, and v) were published in a peer-reviewed journal or as a pre-print on a digital platform or as a report by a credible organisation. We excluded studies if they met any of the following exclusion criteria: i) reported data that were duplicated from a previously included study, ii) no full text available, iii) not available in English. One author screened titles, abstracts and full texts against the inclusion and exclusion criteria. At each stage; title screening, abstract screening and full text review, a random subset (10%) was reviewed by a second author.

We extracted data from eligible studies using a standardised tool that was piloted on 5 studies (S1 Table). Data was abstracted into a Google Sheets document, which was archived on completion of data extraction [32]. We identified the aims of included studies, for example, whether it was conducted as a survey of small mammal species or specifically to assess the risk of zoonotic disease spillover. we extracted data on study methodology, such as, the number of trap nights, the type of traps used and whether the study attempted to estimate abundance. For studies not reporting number of trap nights we used imputation based on the number of trapped individuals, stratified by the habitat type from which they were obtained. This was performed by multiplying the total number of trapped individuals within that study site by the median trap success for study sites with the same reported habitat type. Stratification was used as trap success varied importantly between traps placed in or around buildings (13%, IQR 6–24%) compared with other habitats (3%, IQR 1–9%)

We also recorded how species were identified within a study and species identification was assumed to be accurate. The number of individuals of these species or genera was extracted with taxonomic names mapped to GBIF taxonomy [33]. We expanded species detection and non-detection records by explicitly specifying non-detection at a trap site if a species was recorded as detected at other trapping locations within the same study.

Geographic locations of trapping studies were extracted using GPS locations for the most precise location presented. Missing locations were found using the National Geospatial-Intelligence Agency GEOnet Names Server [34] based on placenames and maps presented in the study. All locations were converted to decimal degrees. The year of rodent trapping was extracted alongside the length of the trapping activity to understand seasonal representativeness of trapping activity. The habitats of trapping sites were mapped to the IUCN Habitat Classification Scheme (Version 3.1). For studies reporting multiple habitat types for a single trap, trap-line or trapping grid, a higher order classification of habitat type was recorded.

For included studies with available data we extracted information on all microorganisms and known zoonotic pathogens tested and the method used (e.g., molecular or serological diagnosis). Where assays were able to identify the microorganism to species level this was recorded, for non-specific assays higher order attribution was used (e.g., to family level). A broad definition of known zoonotic pathogen was used, a species of microorganism carried by an animal that may transmit to humans and cause illness [35]. We do not include evolved pathogens acquired originally through zoonotic pathways in our definition (i.e., HIV). The

term microorganism is used where either the microorganism is not identified to species level, in which case it remains unclear whether it is a zoonotic pathogen (i.e., Arenaviridae), or the species is not known to be a zoonotic pathogen (i.e., *Candidatus Ehrlichia senegalensis*). We recorded the species of rodent host tested, the number of individuals tested and the number of positive and negative results. For studies reporting summary results all testing data were extracted, this may introduce double counting of individual rodents, for example, if a single rodent was tested using both molecular and serological assays. Where studies reported indeterminate results, these were also recorded.

**2.1.2. Description of included studies.** Out of 4,692 relevant citations, we identified 127 rodent trapping studies (S2 Table). Of these, 55 (43%) were conducted to investigate rodent-borne zoonoses, with the remaining 77 (57%) conducted for ecological purposes (i.e., population dynamics, distribution) in rodents, including those known to be hosts of zoonotic pathogens. The earliest trapping studies were conducted in 1964, with a trend of increasing numbers of studies being performed annually since 2000. The median year of first trapping activity was 2007, with the median length of trapping activity being 1 year (IQR 0–2 years) (S1 Fig.). Studies were conducted in 14 West African countries, with no studies reported from The Gambia or Togo, at 1,611 trap sites (Fig 1A.).

Included studies explicitly reported on 601,184 trap nights, a further 341,445 trap nights were imputed from studies with no recording of trapping effort based on trap success, leading to an estimate of 942,629 trap nights (Fig 1B.). A minority of studies trapped at a single study site (30, 24%), with 46 (36%) trapping at between two and five sites, the remaining 51 studies (40%) trapped at between six and 93 study sites.

In total 76,275 small mammals were trapped with 65,628 (90%) identified to species level and 7,439 (10%) identified to genus, with the remaining classified to higher taxonomic level. The majority of the 132 identified species were Rodentia (102, 78%), of which Muridae (73, 72%) were the most common family. Soricomorpha were the second most identified order of small mammals (28, 21%). 57 studies tested for 32 microorganisms, defined to species or genus level that are known or potential pathogens. Most studies tested for a single microorganism (48, 84%). The most frequently assayed microorganisms were *Lassa mammarenavirus* or Arenaviridae (21, 37%), *Borrelia sp.* (9, 16%), *Bartonella sp.* (4, 7%) and *Toxoplasma gondii* (4, 7%). Most studies used Polymerase Chain Reaction (PCR) to detect microorganisms (37, 65%), with fewer studies using serology-based tests (11, 19%) or histological or direct visualisation assays (11, 21%). From 32,920 individual rodent samples we produced 351 host-pathogen pairs. With *Rattus rattus*, *Mus musculus*, *Mastomys erythroleucus*, *Mastomys natalensis* and *Arvicanthis niloticus* being assayed for at least 18 microorganisms.

## 2.2. Analysis

**2.2.1. What is the extent of spatial bias in the rodent trapping data?.** To investigate the extent of spatial bias in the rodent trapping data, we calculated trap-night (TN) density within each West African level-2 administrative region. The sf package in the R statistical language (R version 4.1.2) was used to manipulate geographic data, administrative boundaries were obtained from GADM 4.0.4 [36–38]. Trap-night density ($TN_{density}$) was calculated by dividing the number of trap nights by the area of a level-2 administrative area ($R_{area}$). For studies not reporting trap nights, imputation was used as previously described. Human population density was obtained for the closest year (2005) to the median year of trapping (2007) from Socioeconomic Data and Applications Center (SEDAC) gridded population of the world v4 at ~ 1km resolution [39]. Median population density was then calculated for each level-2 administrative

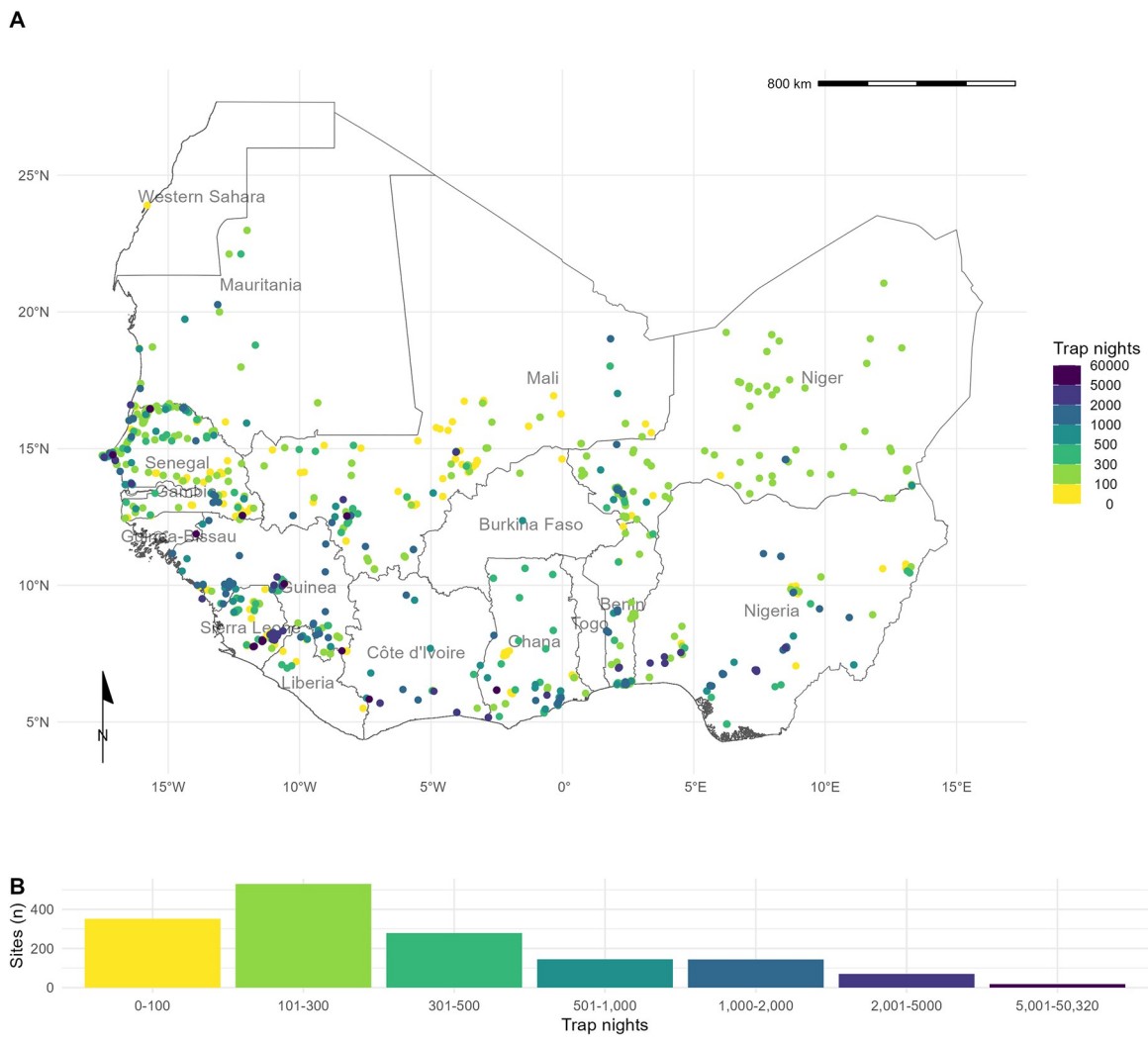

**Fig 1. Rodent trapping sites across West Africa.** A) The location of trapping sites in West Africa. No sites were recorded from Togo or The Gambia. Heterogeneity is observed in the coverage of each country by trap night (colour) and location of sites. For example, Senegal, Mali and Sierra Leone have generally good coverage compared to Guinea and Burkina Faso. B) Histogram of trap nights performed at each study site, a median of 248 trap nights (IQR 116–500) was performed at each site. A labelled map of the study region is attached in S5 Fig. Basemap shapefile obtained from GADM 4.0.4 [38].

region ($P_{density}$). Land cover classification was obtained from the Copernicus climate change service at ~300m resolution [40]. The proportion of cropland, shrubland, tree cover ($\psi_{tree}$) and urban land cover ($\psi_{urban}$) within a level-2 administrative region in 2005 was calculated.

We investigated the association between relative trapping effort, measured as TN density, and the proportion of urban, cropland, tree cover and human population density using Generalised Additive Models (GAM) incorporating a spatial interaction term (longitude and latitude, $X$ and $Y$) [41]. Spatial aggregation of relative trapping effort was modelled using an exponential dispersion distribution (Tweedie) [42]. The models were constructed in the mgcv package [43]. Selection of the most parsimonious model was based on Deviance explained and the Akaike Information Criterion for each model (Eqs 1–5 below). Relative trapping effort was then predicted across West Africa using these covariates. We performed two sensitivity analyses, first, by removing sites with imputed trapping effort, second, by associating trap locations

to ~1km pixels rather than level-2 administrative areas.

$$TN_{density} \sim Tweedie(X * Y) \tag{1}$$

$$TN_{density} \sim Tweedie(P_{density} + (X * Y)) \tag{2}$$

$$TN_{density} \sim Tweedie(P_{density} + R_{area} + (X * Y)) \tag{3}$$

$$TN_{density} \sim Tweedie(P_{density} + \psi_{tree} + \psi_{urban} + (X * Y)) \tag{4}$$

$$TN_{density} \sim Tweedie(P_{density} + R_{area} + \psi_{urban} + (X * Y)) \tag{5}$$

**2.2.2. What is the difference in rodent host distributions between curated datasets and rodent trapping studies?.** We assessed the concordance of curated rodent host distributions from IUCN and GBIF with observed rodent detection and non-detection from rodent trapping studies for seven species with the most trap locations (*M. natalensis, R. rattus, M. erythroleucus, M. musculus, A. niloticus, Praomys daltoni* and *Cricetomys gambianus*). We obtained rodent species distribution maps as shapefiles from the IUCN red list and translated these to a ~20km resolution raster [44]. Distributions were cropped to the study region for globally distributed rodent species. We obtained rodent presence locations from GBIF as point data limited to the study region [45]. Presence locations were associated to cells of raster with a ~20km resolution produced for the study region.

For each of the seven species, we first calculated the area of the IUCN expected range, and then the percentage of this range covered by presence detections in GBIF, and from detections in the rodent trapping data. We then calculated the area of both GBIF and rodent trapping detections outside of the IUCN expected range. For rodent trapping data, we additionally calculated the area of non-detections within the IUCN expected area. Finally, we calculated the combined area of detection from both GBIF and rodent trapping data.

**2.2.3. Are rodent trapping derived host-pathogen associations present in a consolidated zoonoses dataset?.** To examine the usefulness of rodent trapping studies as an additional source of data we compared identified host-pathogen associations from trapping studies investigating zoonoses with a consolidated zoonoses dataset (CLOVER) [11,46]. CLOVER is a synthesis of four host-pathogen datasets (GMPD2, EID2, HP3 and Shaw, 2020) and was released in 2021, it contains more than 25,000 host-pathogen associations for Bacteria, Viruses, Helminth, Protozoa and Fungi. We compared the host-pathogen networks across the two datasets, where the CLOVER data was subset for host species present in the rodent trapping data.

For host-pathogen pairs with assay results consistent with acute or prior infection, we calculated the proportion positive and identify those absent from CLOVER. We expand the analysis to host-pathogen pairs with pathogens identified to genus level in S4 Fig.

**2.2.4. What is the spatial extent of pathogen testing within host ranges?.** We use the sampled area of three pathogen groups and two pathogens (Arenaviridae, Borreliaceae, Leptospiraceae, *Lassa mammarenavirus* and *Toxoplasma gondii*) to quantify the bias of sampling within their hosts ranges. For each pathogen, we first describe the number of host species assayed, for the five most commonly tested species we associate the locations of sampled individuals to ~20km pixels and calculate the proportion of the IUCN range of the host in which sampling has occurred. We compare this figure to the total area in which the host has been detected to produce a measure of relative completeness of sampling within the included rodent trapping studies.

Data and code to reproduce all analyses are available in an archived Zenodo repository [32].

## 3. Results

### 3.1. What is the extent of spatial bias in the rodent trapping data?

We found non-random, spatial clustering of rodent trapping locations across the study region, suggestive of underlying bias in the sampling of rodents across West Africa. Trap sites were situated in 256 of 1,450 (17.6%) level-2 administrative regions in 14 West African nations. The regions with the highest TN density included the capitals and large cities of Niger (Niamey), Nigeria (Ibadan), Ghana (Accra), Senegal (Dakar), and Benin (Cotonou). Outside of these cities, regions in, Northern Senegal, Southern Guinea, Edo and Ogun States in Nigeria and Eastern Sierra Leone had the greatest TN density (Fig 1A.).

The most parsimonious GAM model (adjusted R2 = 0.3, Deviance explained = 48.7%) reported significant non-linear associations between relative trapping effort bias and human population densities (Effective Degrees of Freedom (EDF) = 7.13, $p < 0.001$), proportion of urban landscape (EDF = 1.92, $p < 0.002$) and region area (EDF = 3.63, $p < 0.001$), alongside significant spatial associations (EDF = 27.3, $p < 0.001$) (Supplementary table 3.1). Greatest trapping effort bias peaked at population densities between 5,000–7,500 individuals/km$^2$, proportion of urban landscape >10% and region areas < 1,000km$^2$. Increased trapping effort was found in North West Senegal, North and East Sierra Leone, Central Guinea and coastal regions of Nigeria, Benin and Ghana; in contrast South East Nigeria, Northern Nigeria and Burkina Faso had an observed bias towards a reduced trapping effort (Fig 2). In sensitivity analysis, excluding sites with imputed trap nights, Mauritania, Northern Senegal and Sierra Leone remained as regions trapped at higher rates, with Nigeria being trapped at lower than expected rates (S3A Fig.). In pixel-based sensitivity analysis spatial coverage was reduced with similar patterns of bias observed to the primary analysis (S3B Fig.).

### 3.2. What is the difference in rodent host distributions between curated datasets and rodent trapping studies?

We found that for six of the seven most frequently detected rodent species (*M. natalensis*, *R. rattus*, *M. erythroleucus*, *M. musculus*, *A. niloticus* and *P. daltoni*), trapping studies provided more distinct locations of detection and non-detection than were available from GBIF. For the endemic rodent species (*M. natalensis*, *M. erythroleucus*, *A. niloticus*, *P. daltoni* and *C. gambianus*) IUCN ranges had good concordance to both trapping studies and GBIF, however, individuals of *A. niloticus* and *P. daltoni* were detected outside of IUCN ranges. In contrast, the non-native species *R. rattus* and *M. musculus* were detected across much greater ranges than were expected from IUCN distributions. Comparisons for *M. natalensis*, *R. rattus* and *M. musculus* are shown in Fig 3, the remaining species are shown in S4 Fig.

Comparison of the proportion of a species IUCN range in which detections and non-detections occurred showed that sampling locations of these seven species within GBIF covered between 0.09–0.26% of expected ranges (Table 1.), compared to 0.03–0.24% for rodent trapping data. Detections occurred outside IUCN ranges for all species in both the GBIF and rodent trapping data, most noticeably for *A. niloticus* and *R. rattus*. Combining GBIF and rodent trapping data increased the sampled area by a mean of 1.6 times compared to the GBIF area alone, demonstrating limited overlap between the locations providing information to either dataset. Non-detection of a species occurred across species ranges (mean = 0.11%, SD = 0.03%), suggestive of spatial heterogeneity of presence within IUCN ranges.

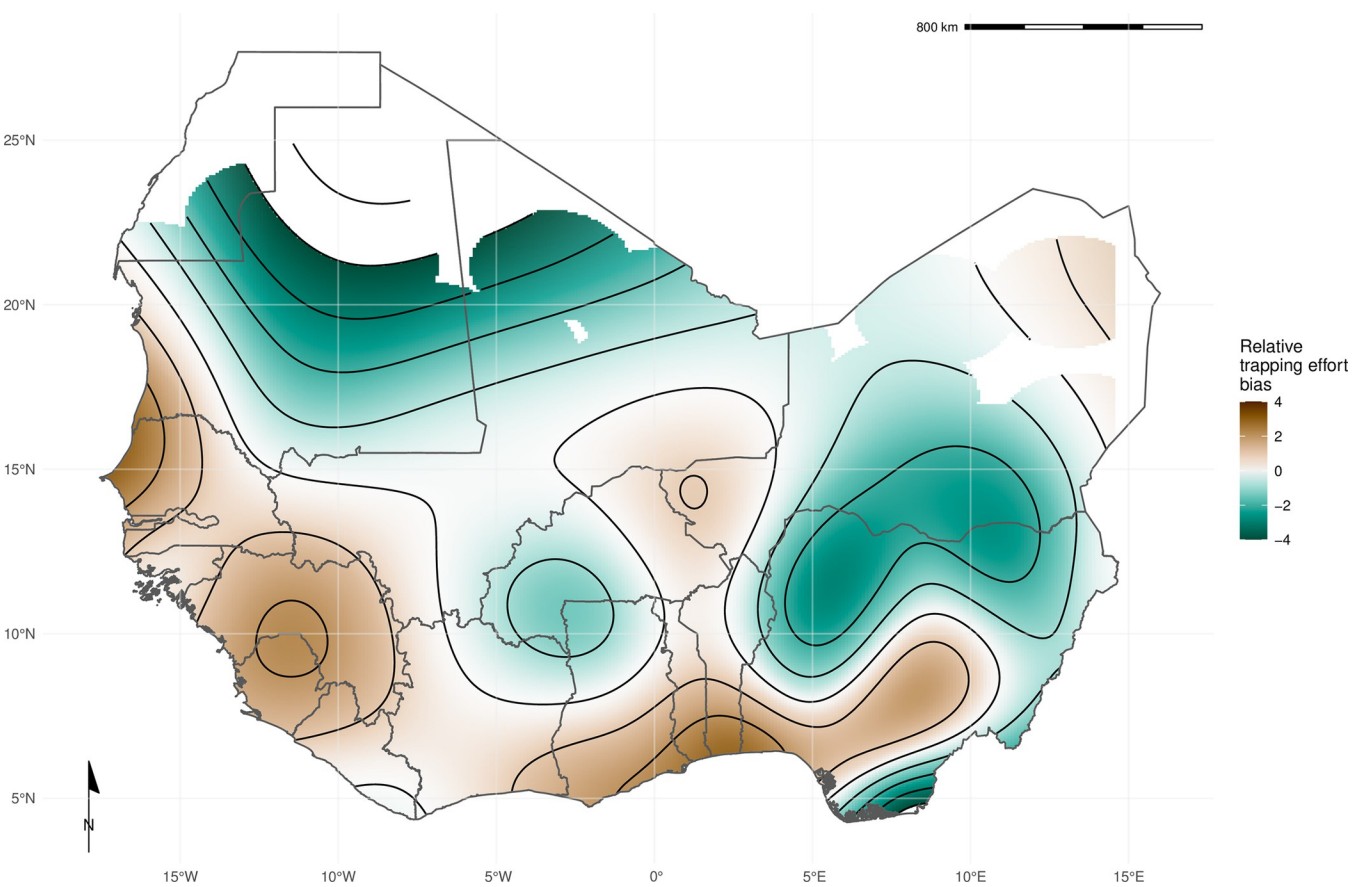

**Fig 2. Relative trapping effort bias across West Africa.** Modelled relative trapping effort bias adjusted for human population density, proportion urban land cover and area of the administrative region. Brown regions represent areas with a bias towards increased trapping effort (e.g., North West Senegal), Green regions represent areas with a bias towards reduced trapping effort (e.g., Northern Nigeria). Basemap shapefile obtained from GADM 4.0.4 [38].

### 3.3. Are rodent trapping derived host-pathogen associations present in a consolidated zoonoses dataset?

We found potentially important differences between the host-pathogen networks produced from included rodent trapping studies and the consolidated CLOVER dataset. When limited to taxonomic classification of both pathogen and host to species level we identified 25 host-pathogen pairs among 14 rodent and 6 pathogen species (Figs 4 and 5). We identified negative associations (non-detection through specific assays) for 45 host-pathogen pairs among 35 rodent and 7 pathogen species. CLOVER contained 10 (40%) of our identified host-pathogen associations, the remaining 15 (60%) were not found to be present in CLOVER, additionally CLOVER recorded positive associations for 4 (9%) of the negative associations produced from the rodent trapping data.

CLOVER included an additional 492 host-pathogen associations we do not observe in rodent trapping studies. The majority of these 392 (80%) pairs are from species with global distributions (*M. musculus*, *R. rattus* and *R. norvegicus*), or from those with wide ranging distributions in sub-Saharan Africa (38, 8%) (i.e., *A. niloticus*, *M. natalensis* and *Atelerix albiventris*).

For pathogens not identified to species level (i.e. family or higher taxa only), we identified 148 host-pathogen pairs among 32 rodent species and 25 pathogen families (S4 Fig.), with CLOVER containing 66 (45%) of these associations.

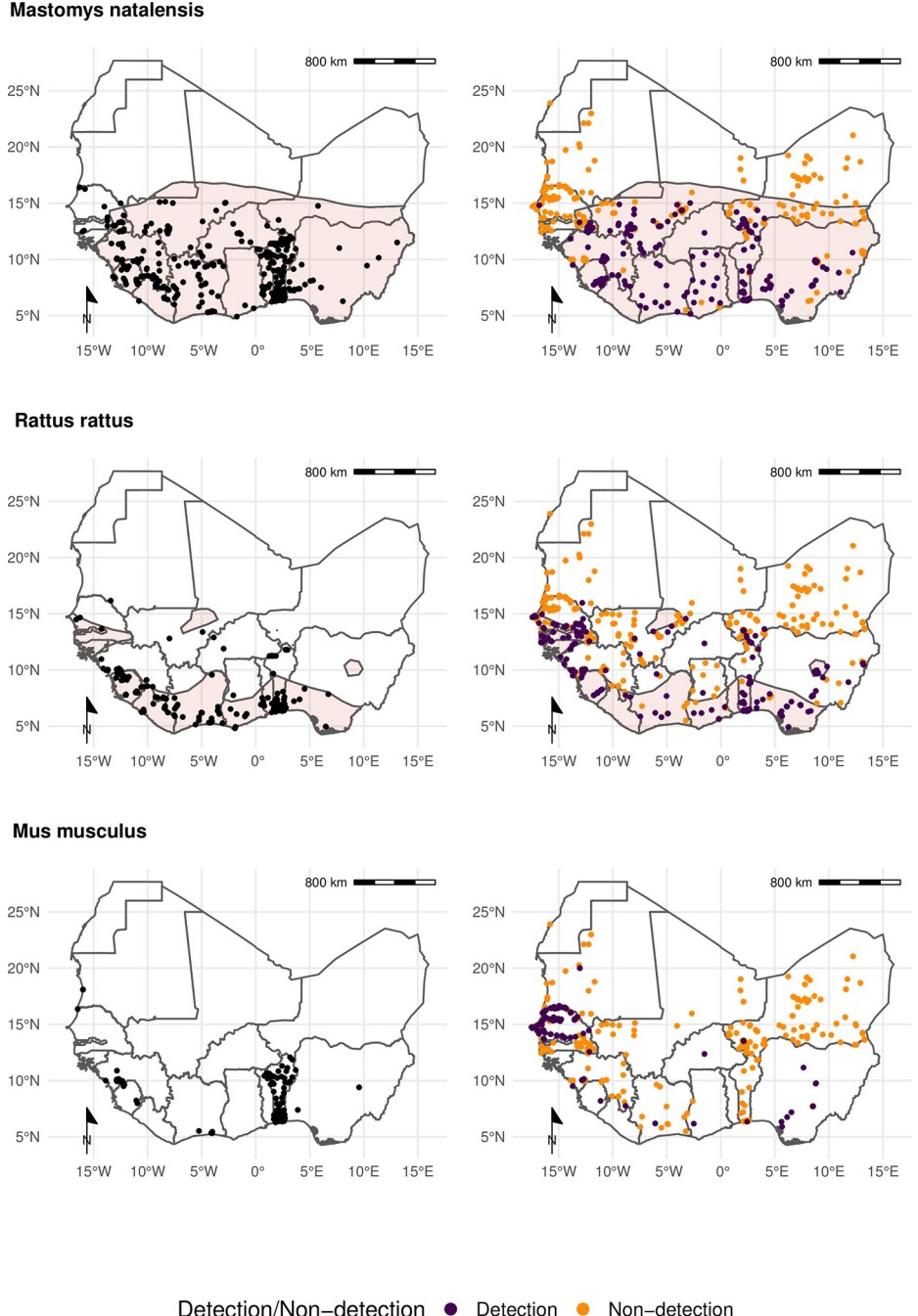

Detection/Non−detection   ● Detection   ● Non−detection

**Fig 3. Locations of detection and non-detection sites for rodent species in West Africa Each row corresponds to a single rodent species.** L) Presence recorded in GBIF (black points) overlaid on IUCN species range (red-shaded area). R) Detection (purple) and non-detection (orange) from rodent trapping studies overlaid on IUCN species ranges. M. musculus has no IUCN West African range. Basemap shapefile obtained from GADM 4.0.4 [38].

Rodent trapping studies identified additional rodent host species for six pathogens; *Lassa mammarenavirus* (5), *Toxoplasma gondii* (4), Usutu virus (2), *Coxiella burnetii* (2), *Escherichia coli* and *Klebsiella pneumoniae* (both 1), that were not present in this consolidated host-pathogen association dataset.

**Table 1. Comparison of IUCN, GBIF and rodent trapping ranges for the 7 most detected rodent species.**

| Species | IUCN | GBIF | | Trapping studies | | | Combined |
|---|---|---|---|---|---|---|---|
| | Range (1,000 km$^2$) | Area inside range (1,000 km$^2$) (% of IUCN range) | Area outside range (1,000 km$^2$) | Detection area inside range (1,000 km$^2$) (% of IUCN range) | Species | Range (1,000 km$^2$) | Area inside range (1,000 km$^2$) (% of IUCN range) |
| *Mastomys natalensis* | 3,257 | 6.83 (0.21%) | 0.19 | 4.4 (0.14%) | *Mastomys natalensis* | 3,257 | 6.83 (0.21%) |
| *Rattus rattus* | 1,019 | 2.61 (0.26%) | 0.52 | 2.42 (0.24%) | *Rattus rattus* | 1,019 | 2.61 (0.26%) |
| *Mastomys erythroleucus* | 3,735 | 4.48 (0.12%) | 0.04 | 3.24 (0.09%) | *Mastomys erythroleucus* | 3,735 | 4.48 (0.12%) |
| *Mus musculus* | | | 2.15 | | *Mus musculus* | | |
| *Arvicanthis niloticus* | 1,829 | 1.69 (0.09%) | 2.41 | 1.98 (0.11%) | *Arvicanthis niloticus* | 1,829 | 1.69 (0.09%) |
| *Praomys daltoni* | 2,658 | 4.03 (0.15%) | 0.29 | 2.03 (0.08%) | *Praomys daltoni* | 2,658 | 4.03 (0.15%) |
| *Cricetomys gambianus* | 2,476 | 5 (0.2%) | 0.17 | 0.75 (0.03%) | *Cricetomys gambianus* | 2,476 | 5 (0.2%) |

### 3.4. What is the spatial extent of microorganism testing within a host's range?

The five most widely sampled microorganism species/families in included studies were Arenaviridae, Borreliaceae, *Lassa mammarenavirus*, Leptospiraceae and *Toxoplasma gondii* (Table 2.). Assays to identify Arenaviridae infection were performed in 44 rodent species with evidence of viral infection in 15 species. Studies that reported Arenaviridae infection did not identify the microorganism to species level and were distinct from those reporting *Lassa mammarenavirus* infection. *Lassa mammarenavirus* was specifically tested for in 43 species with 10 showing evidence of viral infection. The most commonly infected species for both Arenaviridae, generally, and *Lassa mammarenavirus* specifically, were *M. natalensis* and *M. erythroleucus*. These species were assayed across between 10–20% of their trapped area, equating to ~0.02% of their IUCN range (Table 2.).

Infection with species of Borreliaceae was assessed in 42 species, with evidence of infection in 17 rodent species. The greatest rates of infection were among *A. niloticus* (16%), *Mastomys huberti* (11%) and *M. erythroleucus* (9%). Testing was more widespread than for Arenviruses with coverage between 15–34% of their trapped area, however, this remains a small area in relation to their IUCN ranges (<0.05%). Leptospiraceae and *Toxoplasma gondii* was assessed in 8 species, with evidence of infection in 5 and 6 rodent species respectively. The spatial coverage of testing for these microorganisms was more limited within IUCN host species ranges (~0.01%).

### 4. Discussion

Endemic rodent zoonoses and novel pathogen emergence from rodent hosts are predicted to have an increasing burden in West Africa and globally [10]. Here we have synthesised data from 126 rodent trapping studies containing information on more than 72,000 rodents, from at least 132 species of small mammals (Rodentia = 102, Soricidae = 28, Erinaceidae = 2), across 1,611 trap sites producing an estimated 942,669 trap nights from 14 West African countries. Locations studied are complementary to curated datasets (e.g., IUCN, GBIF), incorporation of our synthesised dataset when assessing zoonosis risk based on host distributions could counteract some of the biases inherent to these curated datasets [18]. Most assayed rodents were not found to be hosts of known zoonotic pathogens. We identified 25 host-pathogen pairs reported from included studies, 15 of these were not included in a consolidated host-pathogen

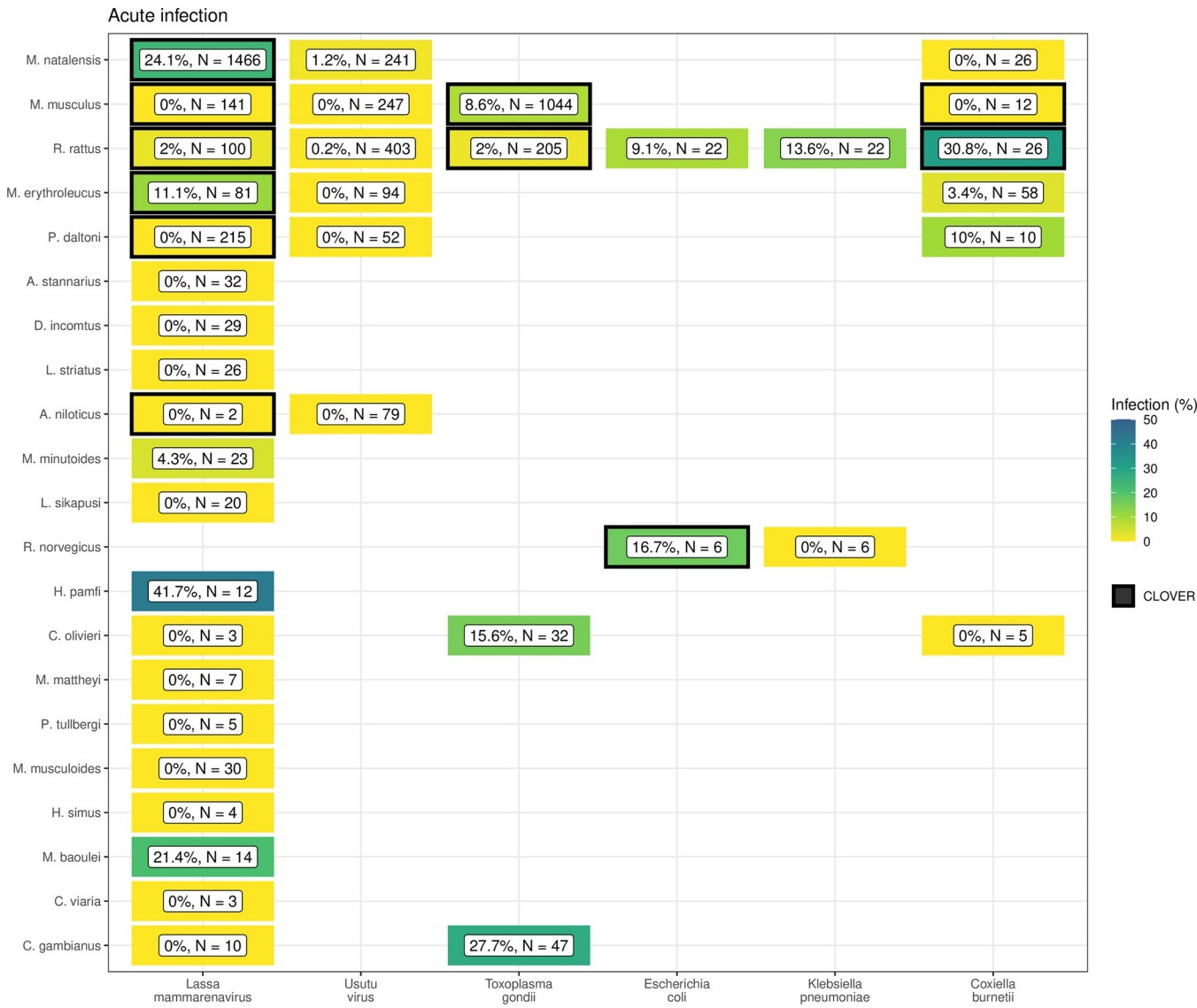

**Fig 4. Host-Pathogen associations detected through acute infection.** A) Identified species level host-pathogen associations through detection of acute infection (i.e. PCR, culture). Percentages and colour relate to the proportion of all assays that were positive, the number of individuals tested for the pathogen is labelled N. Associations with a black border are present in the CLOVER dataset.

dataset. Generally, the number of different species tested for a microorganism and the spatial extent of these sampling locations were limited. These findings highlight a number of sampling bias, supporting calls for further microorganism sampling across diverse species in zoonotic hotspots [47].

We found that rodent trapping data, like biodiversity data, showed important spatial biases [20]. Relative trapping effort bias was greater in Benin, Guinea, Senegal and Sierra Leone driven by long-standing research collaborations investigating the invasion of non-native rodent species (*M. musculus* and *R. rattus*) and the hazard of endemic zoonosis outbreaks (e.g., *Lassa mammarenavirus*). In addition to identifying point locations of prior rodent and pathogen sampling (Fig 1.), additional information on the trapping effort (density of trap-nights), human population density and land use type have been incorporated to produce a value of relative effort that will assist researchers in identifying specific locations where

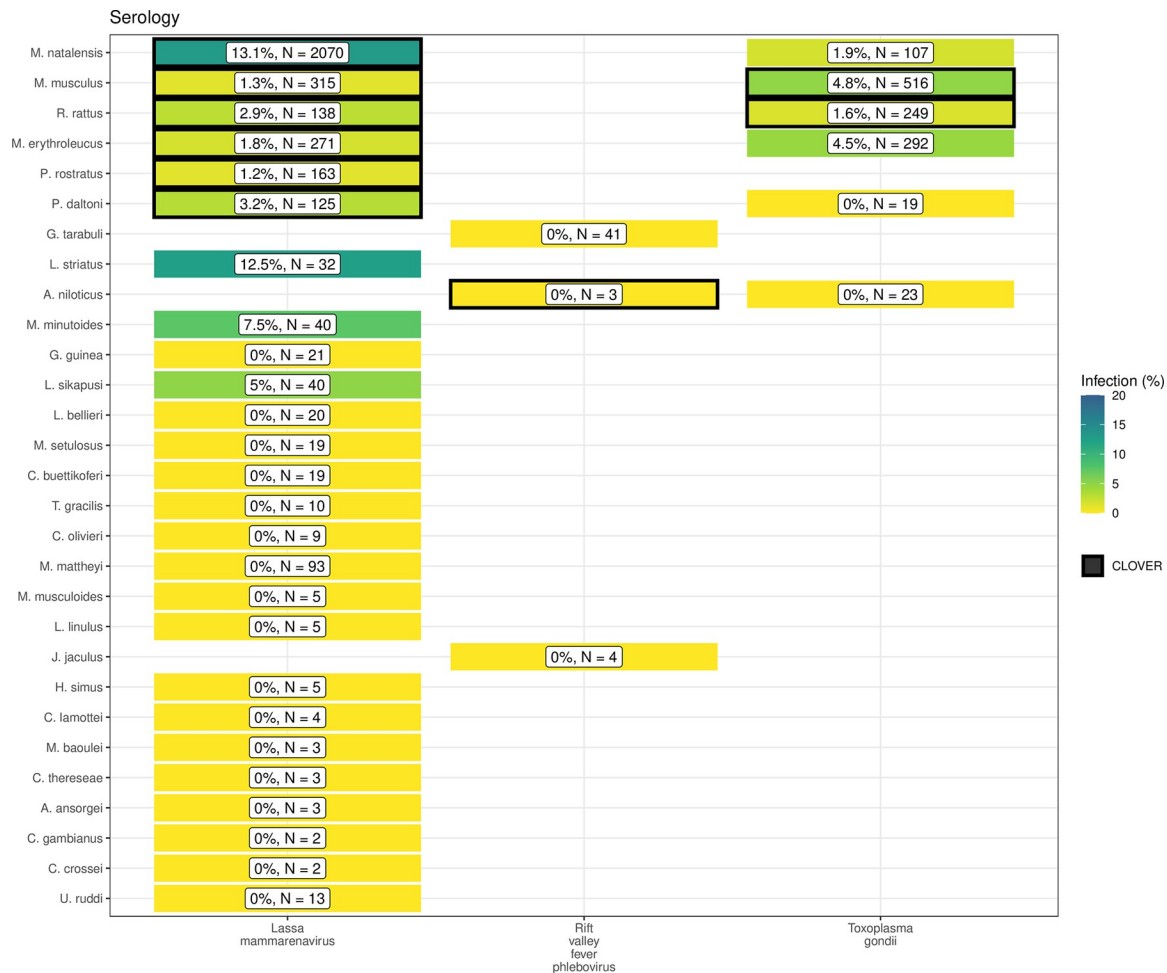

**Fig 5. Host-Pathogen associations detected through evidence of prior infection.** B) Identified species level host-pathogen associations through serological assays (i.e. ELISA). Percentages and colour relate to the proportion of all assays that were positive, the number of individuals tested for the pathogen is labelled N. Associations with a black border are present in the CLOVER dataset.

predictions based on these underlying data sources may suffer from effects of sampling bias. This approach improves the ease of identifying under sampled locations, for example, Fig 1. may suggest that South East Senegal, Southern Mali and Southern Niger are well sampled based on locations of trapping sites. When the number of trap nights, human population density and land use of these regions are taken into account (Fig 2.) and compared with better sampled locations (i.e., Western Senegal, Eastern Sierra Leone) these areas are found to be relatively under sampled and would benefit from further sampling effort. This contrasts to North West Nigeria where no trapping has occurred (Fig 1.), our modelling approach has perhaps highlighted this region as an immediate priority for sampling of rodents and their pathogens given high human population densities and a human dominated landscape.

Much of West Africa remains relatively under sampled, particularly Burkina Faso, Côte d'Ivoire, Ghana and Nigeria, despite these countries facing many of the same challenges. For example, annual outbreaks of Lassa fever are reported in Nigeria and there are potentially 60,000 unrecognised cases of Lassa fever every year in Côte d'Ivoire and Ghana [48]. Our estimates of the proportion of a rodent species range that have been sampled, along with pathogen testing within their sampled range, are sensitive to our choice of raster cell size. Smaller area

**Table 2. Comparison of microorganism sampling ranges for the 5 most widely sampled microorganisms and the 5 most sampled rodent host species (\* no IUCN range in West African).**

| Microorganism | Host species | Tested (N) | Positive (N (%)) | Microorganism testing area (1,000 km²) | Microorganism testing area within trapped area (%) | Microorganism testing area within IUCN range (%) |
|---|---|---|---|---|---|---|
| Arenaviridae sp. | | | | | | |
| | *Mastomys natalensis* | 2,841 | 104 (4%) | 0.61 | 13.45% | 0.02% |
| | *Praomys daltoni* | 854 | 6 (1%) | 0.42 | 19.43% | 0.02% |
| | *Mastomys erythroleucus* | 398 | 20 (5%) | 0.40 | 11.97% | 0.01% |
| | *Rattus rattus* | 396 | 4 (1%) | 0.38 | 10.5% | 0.04% |
| | *Praomys rostratus* | 310 | 5 (2%) | 0.13 | 12.53% | 0.02% |
| Borrelia sp. | | | | | | |
| | *Mastomys erythroleucus* | 1,586 | 140 (9%) | 1.14 | 33.94% | 0.03% |
| | *Arvicanthis niloticus* | 1,551 | 253 (16%) | 0.66 | 28.48% | 0.03% |
| | *Mastomys natalensis* | 733 | 54 (7%) | 0.69 | 15.08% | 0.02% |
| | *Mastomys huberti* | 731 | 83 (11%) | 0.23 | 29.83% | 0.04% |
| | *Mus musculus* | 686 | 26 (4%) | 0.45 | 24.54% | * |
| Lassa mammarenavirus | | | | | | |
| | *Mastomys natalensis* | 3,199 | 580 (18%) | 1.03 | 22.65% | 0.03% |
| | *Mastomys erythroleucus* | 352 | 14 (4%) | 0.36 | 10.63% | 0.01% |
| | *Rattus rattus* | 177 | 2 (1%) | 0.34 | 9.26% | 0.03% |
| | *Praomys rostratus* | 163 | 2 (1%) | 0.27 | 27.02% | 0.04% |
| | *Mus musculus* | 147 | 0 (0%) | 0.04 | 2.29% | * |
| Leptospira sp. | | | | | | |
| | *Rattus rattus* | 646 | 65 (10%) | 0.40 | 11.1% | 0.04% |
| | *Arvicanthis niloticus* | 221 | 10 (5%) | 0.02 | 0.9% | <0.01% |
| | *Crocidura olivieri* | 141 | 14 (10%) | 0.34 | 25.16% | * |
| | *Mastomys natalensis* | 136 | 26 (19%) | 0.36 | 7.91% | 0.01% |
| | *Rattus norvegicus* | 79 | 19 (24%) | 0.21 | 40.08% | * |
| Toxoplasma gondii | | | | | | |
| | *Mus musculus* | 1,548 | 115 (7%) | 0.62 | 33.64% | * |
| | *Rattus rattus* | 428 | 8 (2%) | 0.36 | 9.77% | 0.03% |
| | *Mastomys erythroleucus* | 292 | 13 (4%) | 0.37 | 11.06% | 0.01% |
| | *Mastomys natalensis* | 107 | 2 (2%) | 0.08 | 1.83% | <0.01% |
| | *Cricetomys gambianus* | 47 | 13 (28%) | 0.06 | 7.6% | <0.01% |

cells will reduce the reported coverage while larger cells will have the opposite effect. Despite this, the observed patterns are unlikely to importantly change, with the finding of sparse sampling of both rodents and their pathogens remaining present across cell scales. Rodent sampling should be targeted towards currently under sampled regions to reduce the potential impact of current biases and improve our understanding of both the distribution of rodent

hosts and the prevalence of pathogens within their populations. This will allow for better estimation of risk from endemic and novel zoonoses.

Rodent trapping studies provide geographic and temporally contextualised data on both species detection and non-detection which are not available from curated datasets. Non-detection data can improve models of species distributions, unfortunately, high levels of missing data on trapping effort will continue to confound the allocations of non-detections as true absences [49]. Models of host species occurrence and abundance, improved by incorporating species absence, are important to assess the effect of land use and climate change on endemic zoonosis spillover to human populations and direct limited public health resources towards regions at greatest risk [50,51].

Currently available consolidated datasets on host-pathogen associations (e.g., CLOVER, EID2 and GMPD2) do not include spatial or temporal components [52]. The current synthesis of rodent trapping studies has highlighted that pathogens have been sparsely sampled within a host's range. Current zoonosis risk models dependent on these sources of data are therefore not able to incorporate spatial heterogeneity in pathogen prevalence across the host range. Additional uncertainty in current models of zoonotic disease risk arises from host-pathogen associations that have not been reported in these consolidated datasets. For example, *Hylomyscus pamfi* infected with *Lassa mammarenavirus* and *R. rattus* infected with *Coxiella burnetii*, will not be included when solely based on consolidated host-pathogen datasets. Further, detection of zoonotic pathogens in multiple, co-occurring, host species supports the adoption of multi-species approach to better understand the potential range of endemic zoonoses [53].

Few studies stratified detection and non-detection of hosts or pathogen prevalence by time, therefore limiting inference of temporal changes in host and pathogen dynamics. This limitation prevents calculation of incidence of infection and the abundance of infectious rodents which potentially varies by both time and space [54]. Understanding temporal changes in viral burden and shedding for endemic zoonoses is required to accurately predict current and future risk of pathogen spillover.

Finally, due to data sparsity, we were unable to account for temporal change over the six decades of rodent trapping studies. Land use change and population density have changed dramatically over this period in West Africa [55]. We attempted to mitigate against this by using the median year of trapping to understand the spatial and land use biases in trapping activity. It is possible that land use and population density at trapping sites varied importantly between when rodent trapping was conducted and the conditions in 2005. Despite this limitation, the finding that trapping is biased towards high density, human dominated landscapes is unlikely to substantially change.

We have shown that synthesis of rodent trapping studies to supplement curated rodent distributions can counteract some of the inherent biases in these data and that they can add further contextual data to host-pathogen association data. Together this supports their inclusion in efforts to model endemic zoonotic risk and novel pathogen emergence. Contribution of rodent trapping studies as data sources can be improved by adopting reporting standards and practices consistent with Open Science, namely sharing of disaggregated datasets alongside publication [56].

Future rodent trapping studies should be targeted towards regions that are currently understudied. Further information on rodent presence and abundance across West Africa will aid the modelling of changing endemic zoonosis risk and the potential for novel pathogen emergence. Sharing of disaggregated data alongside research publications should be promoted with adoption of data standards to support ongoing data synthesis. Specifically, inclusion of exact locations of trapping sites, trapping effort and the dates at which trapping occurred would support more detailed inference of the spatio-temporal dynamics of host populations and the risk

of endemic zoonosis spillover events. Despite these challenges we propose that rodent trapping studies can provide an important source of data to supplement curated datasets on rodent distributions to quantify the risk of endemic zoonosis spillover events and the hazard of novel pathogen emergence.

## Supporting information

**S1 Table. Data extraction tool for studies meeting inclusion criteria.**
(DOCX)

**S2 Table. Included studies.**
(DOCX)

**S3 Table. GAM outputs for the association of relative trapping effort and covariates of interest.**
(DOCX)

**S1 Fig. Timeline of included studies.** Green points represent the start date of rodent trapping studies, blue points representing the final trapping activity. Red points indicate the publication of studies. Increasing numbers of studies have been published since 2000 with more studies being conducted over repeated visits.
(DOCX)

**S2 Fig. Relative trapping effort bias across West Africa from the subset of included studies reporting trapping effort, adjusted for proportion urban land cover and proportion tree cover.** Brown regions represent areas with higher than expected trapping effort, green regions represent areas lower than expected trapping effort. Basemap shapefile obtained from GADM 4.0.4 [38].
(DOCX)

**S3 Fig. Pixel based analysis of relative trapping effort bias across West Africa adjusted for habitat type and human population density.** Brown regions represent areas with higher than expected trapping effort, green regions represent areas lower than expected trapping effort. Basemap shapefile obtained from GADM 4.0.4 [38].
(DOCX)

**S4 Fig.** A) Identified host-pathogen associations at pathogen family level through detection of acute infection (i.e. PCR, culture). B) Identified host-pathogen associations at pathogen family level through serological assays (i.e. ELISA). Percentages and colour relate to the proportion of all assays that were positive. Associations with a black border are present in the CLOVER dataset.
(DOCX)

**S5 Fig. A map of the study region with capital cities and areas discussed in the manuscript highlighted.** Basemap shapefile obtained from GADM 4.0.4 [38].
(DOCX)

**S6 Fig. Locations of detection and non-detection sites for rodent species in West Africa.** Each row corresponds to a single rodent species. L) Presence recorded in GBIF (black points) overlaid on IUCN species range (red-shaded area). R) Detection (purple) and non-detection (orange) from rodent trapping studies overlaid on IUCN species ranges. M. musculus has no IUCN West African range. Basemap shapefile obtained from GADM 4.0.4 [38].
(DOCX)

## Author Contributions

**Conceptualization:** David Simons, Kate E. Jones.

**Data curation:** David Simons, Lauren A. Attfield.

**Formal analysis:** David Simons.

**Funding acquisition:** Deborah Watson-Jones, Richard Kock.

**Supervision:** Kate E. Jones, Deborah Watson-Jones, Richard Kock.

**Validation:** Lauren A. Attfield.

**Writing – original draft:** David Simons.

**Writing – review & editing:** Lauren A. Attfield, Kate E. Jones, Deborah Watson-Jones, Richard Kock.

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
