## [Decision Letter · Decision Letter 0]

21 Oct 2022

Dear Dr Simons,

Thank you very much for submitting your manuscript "Rodent trapping studies as an overlooked information source for understanding endemic and novel zoonotic spillover." for consideration at PLOS Neglected Tropical Diseases. As with all papers reviewed by the journal, your manuscript was reviewed by members of the editorial board and by several independent reviewers. In light of the reviews (below this email), we would like to invite the resubmission of a significantly-revised version that takes into account the reviewers' comments. 

Your manuscript has been reviewed by three experts, all of whom concluded that the studies you describe are highly relevant to the study and control of zoonotic disease and were well executed. With some specific exceptions, they also thought the manuscript was well written and presented. Each reviewer did however have comments and queries for you to address relative to procedures and conclusions. We request that you respond to these reviewer comments and modify your manuscript accordingly.

We cannot make any decision about publication until we have seen the revised manuscript and your response to the reviewers' comments. Your revised manuscript is also likely to be sent to reviewers for further evaluation.

Sincerely,

Richard A. Bowen

Academic Editor

Dileepa Ediriweera

Section Editor

Your manuscript has been reviewed by three experts, all of whom concluded that the studies you describe are highly relevant to the study and control of zoonotic disease and were well executed. With some specific exceptions, they also thought the manuscript was well written and presented. Each reviewer did however have comments and queries for you to address relative to procedures and conclusions. We request that you respond to these reviewer comments and modify your manuscript accordingly.

Reviewer's Responses to Questions

**Key Review Criteria Required for Acceptance?**

**Methods**

-Are the objectives of the study clearly articulated with a clear testable hypothesis stated?

-Is the study design appropriate to address the stated objectives?

-Is the population clearly described and appropriate for the hypothesis being tested?

-Is the sample size sufficient to ensure adequate power to address the hypothesis being tested?

-Were correct statistical analysis used to support conclusions?

-Are there concerns about ethical or regulatory requirements being met?

Reviewer #1: -Are the objectives of the study clearly articulated with a clear testable hypothesis stated?

Yes

-Is the study design appropriate to address the stated objectives?

Yes but requires at least one reviewer who is a specialist of the ecological analysis.

-Is the population clearly described and appropriate for the hypothesis being tested?

Yes

-Is the sample size sufficient to ensure adequate power to address the hypothesis being tested?

Yes

-Were correct statistical analysis used to support conclusions?

I assume yes but need another reviewer in the ecology field.

-Are there concerns about ethical or regulatory requirements being met?

No concerns

Reviewer #2: The objectives of the study are clearly stated, and study methods are appropriate to address these, with an extensive literature search conducted to inform the analyses.

Reviewer #3: Methods

General comment: The methods section is clearly written and provides enough detail to follow the experiment. However, the presented study includes data from trapping studies datingback to the 1960ties. However, it is questionable if the inclusion period of the studies over 60 years is really adding validity to the author’s conclusions– since, and as the authors also state – the human population density and distribution, land use and climate changed drastically during this 6 decades in West Africa, and hence also the composition of the rodent fauna in the area most likely changed. Therefore, it would be worth to reconsider limiting this time span to max three decades to increase significance of the presented work. Additionally, it would be an added value to more prominently highlight the number of studies that include information on actual known rodent species of concern here, namely those important for the zoonotic diseases reported on in this manuscript, and considering limiting the exercise to these studies.

Line 164 Here, action is needed by adding clear information on what the authors refer to as “relevant studies”.

Lines 164-171 From reading this paragraph, it is not clear on what basis and to what extend any “microorganisms and zoonotic pathogens” were included, and how “all microorganisms tested” are relevant for a manuscript reporting on zoonotic disease spill over. The authors are encouraged to revise to allow for more clarity and easier understanding for the reader. 

Line 172-182 This paragraph reports on the data sources as identified through the review process and includes also certain analysis and results – therefore consider a separate heading for this paragraph.

**Results**

-Does the analysis presented match the analysis plan?

-Are the results clearly and completely presented?

-Are the figures (Tables, Images) of sufficient quality for clarity?

Reviewer #1: -Does the analysis presented match the analysis plan?

Yes

-Are the results clearly and completely presented?

Yes

-Are the figures (Tables, Images) of sufficient quality for clarity?

Yes

Reviewer #2: The results are clearly presented, and match the analysis plan. Figures and tables are well done and clearly communicate the results.

Reviewer #3: ANALYSIS / RESULTS

Figure 3 The presentation of Mus musculus data rises several questions here. There seems to be a very limited overlap of “detection” data between the GBIF information and the trapping studies. GBIF presents occurrence data covering Benin, while the trapping studies could not detect any Mus musculus in Benin. Can the authors explain this discrepancy?

Table 2 The headings of the columns need attention: please add clear information on what the numbers report (for example the column “Tested” shall include “Tested (N)” etc.)

**Conclusions**

-Are the conclusions supported by the data presented?

-Are the limitations of analysis clearly described?

-Do the authors discuss how these data can be helpful to advance our understanding of the topic under study?

-Is public health relevance addressed?

Reviewer #1: -Are the conclusions supported by the data presented?

Yes

-Are the limitations of analysis clearly described?

Yes

-Do the authors discuss how these data can be helpful to advance our understanding of the topic under study?

Yes absolutely

-Is public health relevance addressed?

Yes

Reviewer #2: In general the conclusions are well-presented and supported by the data. The public health relevance is clearly addressed. A couple of points that could be clarified:

- Line 319- Assume that these coverage values are based on the proportion of raster cells to which a point trapping location was allocated, in which case the values are highly dependent on the raster size chosen, which is a limitation that should be made clear.

- One of the aims is to establish sampling bias in relation to human population density and land use. However, it is not made clear in the conclusions why the predicted trapping effort is of more use use than simply identifying geographic areas with less sampling (e.g. benefits of Figure 2 rather than identifying gaps in Figure 1). E.g. It may be useful to increase sampling in non-urban areas with lower human density, but this is not clearly discussed.

Reviewer #3: General comment:

 The discussion is a great reading and brings to the point a variety of challenges authors faced while performing this nicely designed study.

 However, I encourage the authors to revise it in view of my general comments to the manuscript.

Lines 386-388 In addition to the number of rodents trapped please also provide the information of how many species these belonged to.

**Editorial and Data Presentation Modifications?**

Reviewer #1: In the figure 1A and B, country and city names mentioned in the main text should be indicated for readers who are not familiar with West African countries and their cities.

Reviewer #2: -

Reviewer #3: (No Response)

**Summary and General Comments**

Reviewer #1: The present manuscript submitted by D Simons et al. to the PLOS Neglected Tropical Diseases journal addressed to review and summarize the previously published trapping studies targeting rodents and pathogens they carry in Western African countries. Since the individual studies may not be able to discover anything beyond their study focus and area, I really love to read this kind of studies challenging to extract the gaps of them. The study seems to be firmly performed and presentation and interpretation of data are fair.

A major issue from a virological standing point is that the overlapping between a pathogen group (Arenaviridae family) and a pathogen (Lassa mammareanavirus). Since other three targets in the present manuscript (Borreliaceae, Leptospiraceae, and Toxoplasma gondii) are all independent and not overlapped each other, the incongruity of selection of pathogens and pathogen groups may not be acceptable, especially by virologists. The authors may want to find which arenavirus was detected in the studies detecting Arenaviridae family and divide them into Lassa mammarenavirus and the others.

Minor issue

L. 61. “wildlife defaunation” may not be suitable to be listed here since this should be a result of “intensifying anthropogenic pressure”.

Reviewer #2: This is a well-written and useful synthesis of rodent trapping studies in West Africa, that has identified new host-pathogen associations and potential gaps in our understanding of host and pathogen distributions, with clear public health relevance. 

A few additional minor comments: 

- The introduction could benefit from clarifying the difference in how curated data sets (e.g. GBIF and CLOVER) obtain records and the considered trapping studies. E.g. Are these trapping studies not previously included in curated databases because of data access issues? 

- Line 70- Is 4 days the correct number here? This is too short for a gestation period

- Line 227- May be useful to explain the use of Tweedie here.

Reviewer #3: While the study works on relevant questions, the way it is reported in this manuscript is not suitable for publication. The way we scientists report about zoonotic diseases and the animals affected needs to be carefully revised. As in this manuscript, the authors report about “rodents” in an indiscriminate way – a very diverse and ecologically very important group of animal species, that play an important role in the ecosystem. While only a few rodent species are known to play a role in zoonotic disease transmission, the reaction and actions taken by people based on such generalized ways of communicating information will and are mostly targeting any rodent species, with a huge negative impact on their abundance, diversity and occurrence. However, rodents are much more than “…important globally distributed reservoirs of known and novel zoonotic pathogens”. I am a very strong advocate for change in our human – animal relationship, and here, I see that we scientists have an important role to play, including in the way we communicate. The rodent’s population dynamic is heavily impacted on by habitat changes and landscape modulations caused by humans and this is equally applying for their health, fitness and exposure to pathogens. To achieve health for all – what in my reading includes human, animal, plant and environmental health, we scientist have an important role to play and be sensitive in the way we communicate. Therefore, I encourage the authors to carefully revise their manuscript with a One Health lens to avoid any inappropriate generalization of “rodents” and rather focus on promoting a better understanding of human activities and its impact on animals and zoonotic pathogens.

PLOS authors have the option to publish the peer review history of their article (what does this mean?). If published, this will include your full peer review and any attached files.

Reviewer #1: No

Reviewer #2: No

Reviewer #3: No
---

## [Decision Letter · Decision Letter 1]

11 Dec 2022

Dear Dr Simons,

Thank you very much for submitting your manuscript "Rodent trapping studies as an overlooked information source for understanding endemic and novel zoonotic spillover." for consideration at PLOS Neglected Tropical Diseases. As with all papers reviewed by the journal, your manuscript was reviewed by members of the editorial board and by several independent reviewers. The reviewers appreciated the attention to an important topic. Based on the reviews, we are likely to accept this manuscript for publication, providing that you modify the manuscript according to the review recommendations. 

Three reviewers considered the revision you submitted favorably and that the manuscript is acceptable for publication in PNTD, but have small suggestions for improvement that you may want to consider. Please evaluate those suggestions and, if you agree, please edit the manuscript to reflect those changes, then resubmit.

Sincerely,

Richard A. Bowen

Academic Editor

Dileepa Ediriweera

Section Editor

Reviewer's Responses to Questions

**Key Review Criteria Required for Acceptance?**

**Methods**

-Are the objectives of the study clearly articulated with a clear testable hypothesis stated?

-Is the study design appropriate to address the stated objectives?

-Is the population clearly described and appropriate for the hypothesis being tested?

-Is the sample size sufficient to ensure adequate power to address the hypothesis being tested?

-Were correct statistical analysis used to support conclusions?

-Are there concerns about ethical or regulatory requirements being met?

Reviewer #1: (No Response)

Reviewer #2: (No Response)

Reviewer #3: The authors have addressed my reviewer comments from revision round. No further revisions required.

**Results**

-Does the analysis presented match the analysis plan?

-Are the results clearly and completely presented?

-Are the figures (Tables, Images) of sufficient quality for clarity?

Reviewer #1: (No Response)

Reviewer #2: (No Response)

Reviewer #3: The authors have addressed my reviewer comments from revision round. No further revisions required.

**Conclusions**

-Are the conclusions supported by the data presented?

-Are the limitations of analysis clearly described?

-Do the authors discuss how these data can be helpful to advance our understanding of the topic under study?

-Is public health relevance addressed?

Reviewer #1: (No Response)

Reviewer #2: (No Response)

Reviewer #3: DISCUSSION

Lines 413 - 416:

In response to the point raised during revision round 1, the authors now state that the more than 72k trapped rodents belong to “at least 132 species of small mammals”. The IUCN defines the group of small mammals to comprise rodents, tree shrews and eulipotyphlans – the latter two therewith being non-target species groups for the presented study.

 Here, the authors shall provide clarity if the study included information of non-rodent species as well (including three shrews and eulipotyphlans), or otherwise clearly state how many of the “at least 132 species” were actually rodents and how many non-rodent small mammals.

**Editorial and Data Presentation Modifications?**

Reviewer #1: Since the authors did not modify the figures 1A and B as I suggested to add names of countries and cities "in the figures", I could not judge whether the addition will obscure the data or not. The authors should show the figures which were actually obscured. Addition of the new supplementary figure 5 did not improve the readability than google maps.

Reviewer #2: (No Response)

Reviewer #3: (No Response)

**Summary and General Comments**

Reviewer #1: This reviewer considers that the authors did revisit the manuscript well, and the revised manuscript may be able to publish as the current form except for the point I raised in the Data Presentation Modification. Since the comment I made is not a critical for their study itself, this reviewer is mostly satisfied with the quality of the study.

Reviewer #2: The paper is a useful and informative review of the information provided by rodent trapping studies in West Africa, and the authors have suitably addressed prior comments. My only minor comment is a few sentences could do with rewording for clarity:

e.g. Line 16- needs comma after rodents

Line 18- Composition of these species' abundance does not make sense. Suggest removing "the composition"

Line 20- "demand that a better understanding of the current distribution of rodent species is vital". Suggest rewording.

Reviewer #3: The authors have addressed most of the comments from the first round of revisions and provided most informative and useful answers to the questions raised during revision round 1, and hence, the manuscript has improved considerably. To be suitable for publication, one minor revision remains to be addressed by the authors.

PLOS authors have the option to publish the peer review history of their article (what does this mean?). If published, this will include your full peer review and any attached files.

Reviewer #1: No

Reviewer #2: No

Reviewer #3: No

Figure Files:

Data Requirements:

Reproducibility:

References

---

## [Editor Report · Decision Letter 2]

15 Jan 2023

Dear Dr Simons,

We are pleased to inform you that your manuscript 'Rodent trapping studies as an overlooked information source for understanding endemic and novel zoonotic spillover.' has been provisionally accepted for publication in PLOS Neglected Tropical Diseases.

Best regards,

Richard A. Bowen

Academic Editor

Dileepa Ediriweera

Section Editor

The adjustments you made to your manuscript were viewed quite favorably by the reviewers - thank you for those revisions. Nice job on a very important topic.

---

## [Editor Report · Acceptance letter]

18 Jan 2023

Dear Dr Simons,

We are delighted to inform you that your manuscript, "Rodent trapping studies as an overlooked information source for understanding endemic and novel zoonotic spillover.," has been formally accepted for publication in PLOS Neglected Tropical Diseases.

Best regards,

Shaden Kamhawi

co-Editor-in-Chief

Paul Brindley

co-Editor-in-Chief
